# Simultaneous Measurement of Thickness and Elastic Properties of Thin Plastic Films by Means of Ultrasonic Guided Waves

**DOI:** 10.3390/s21206779

**Published:** 2021-10-13

**Authors:** Rymantas Jonas Kažys, Olgirdas Tumšys

**Affiliations:** Ultrasound Research Institute, Kaunas University of Technology, K. Baršausko St. 59, LT-51423 Kaunas, Lithuania; rymantas.kazys@ktu.lt

**Keywords:** PVC films, ultrasonic guided waves, ultrasonic measurements, plastic materials, elastic properties of PVC film, thickness measurement

## Abstract

Ultrasonic guided waves are already used for material characterization. The advantage of these waves is that they propagate in the plane of a plate and their propagation characteristics are sensitive to properties of the material. The objective of this research was to develop an ultrasonic method that could be used to measure the properties of thin plastic polyvinylchloride films (PVC). The proposed method exploits two fundamental Lamb wave modes, A_0_ and S_0_, for measurement of a thin film thickness and Young’s modulus. The Young’s modulus is found from the measured phased velocity of the S_0_ mode and the film thickness from the velocities of both A_0_ and S_0_ modes. By using the proposed semi-contactless measurement algorithm, the Young’s modulus and thickness of different thickness (150 µm and 200 µm) PVC films were measured. The uncertainty of thickness measurements of the thinner 150 µm PVC film is 2% and the thicker 200 µm PVC film is 3.9%.

## 1. Introduction

Ultrasonic waves are already used to measure the elastic properties and thickness of solid materials. These properties, such as the Young’s modulus and Poisson’s ratio, can easily be estimated by measuring the propagation velocities of longitudinal or shear ultrasonic waves in isotropic materials [1]. For such measurements, bulk samples with much larger dimensions than the ultrasonic wavelength are usually used. In the case of thin plates and films two different approaches may be applied. The first one is based on the application of high frequency ultrasonic waves incident to the front surface of a film and the analysis of the reflected signals. The second method uses guided Lamb or surface acoustic waves that are excited in the plate [2]. The frequency of the used ultrasonic waves as a rule is lower than in the first approach.

Measurement of elastic properties of thin films by high frequency ultrasonic waves was analyzed by a few authors [3,4]. For example, measurement of elastic properties and density of thermoplastic thin polymer films such as polyethylene terephthalate, polycaprolactone triol, and polyvinyl butyral was discussed in [3]. The samples of thin films samples were placed on a solid substrate and immersed in water. Measurements were performed by a pulse-echo method using high frequency 50 MHz plane ultrasonic waves reflected from the thin films. The elastic moduli and density were found solving the inverse problem using the reflected waves. The obtained values of mechanical parameters correspond to the values reported by other researchers. However, measurement of the thickness of such films was not discussed.

Application for measurements of a focused ultrasonic beam was discussed in [4]. In this paper, the authors analyzed a double focus method based on irradiation of thin metallic plates by a high frequency 57 MHz ultrasonic focused beam. The depth scanning of samples changing the distance between a focused ultrasonic transducer was performed. The focused signals reflected from front and back surfaces of the plate were used for measurements. For a better understanding of the obtained results, the authors proposed the ray model of multi-mode focusing. Experiments were performed in thin stainless steel and aluminium alloy plates immersed in water. The authors claim that the proposed multi-model method can be used for simultaneous sound-velocity and thickness measurement if the density of the material is known, however this topic is not elaborated.

Such methods have several limitations. The wavelength must be much smaller than the thickness of the sample what in the case of thin films requires very high frequencies of ultrasonic waves. Attenuation of ultrasonic waves increases with frequency and it can reduce the accuracy of measurements. Samples must be immersed in water, which in some cases is not allowed. The parameters of the samples are measured in the thickness direction at the selected position; therefore, in order to get a spatial distribution of thickness, the C-scan of the sample is required, etc. [2].

Some of those limitations can be avoided using the guided Lamb waves instead of bulk waves. The advantage of those waves is that they propagate in the plane of a plate and their propagation characteristics are sensitive to properties of the material through which they propagate [5,6,7,8]. On the other hand, the Lamb waves have a multimode and dispersive nature.

Nowadays, ultrasonic guided waves are widely used not only for detection of various defects in planar structures [5,6,7], but for material characterization purposes also. Such characterization is directed to determination of some elastic properties and a thickness of the sample under testing [7,8,9,10,11,12].

Determination of elastic properties and thickness of thin plates and films usually is based on measurement of propagation velocities of at least two different guided wave modes, for example A_0_ and S_0_ modes, and reconstruction of those parameters from the measured velocities [7,8,9,10,11,12,13,14]. For reconstruction, the theoretical velocity dispersion curves are adjusted to experimental results, and in such way the elastic constants and/or thickness of the thin planar specimen are obtained.

Propagation of Lamb waves in a thin gold film on a fused silica semi-infinite substrate was analyzed in [8]. In the presented research, a numerical inversion procedure was used to determine elastic properties of the gold layer. For measurements, a scanning acoustic microscopy method was used. It was found that the determined Poisson’s coefficient and the density of the gold film were slightly lower and the Young’s modulus was higher than that of the bulk gold material.

A method based on fitting of the theoretical Lamb waves dispersion curves to experimental data is presented in [9]. This paper describes determining of the thickness and two independent elastic constants of aluminium plates of a few millimetres thickness. Lamb wave modes at different frequencies are generated by means of the contact prismatic coupling block method. The visualization of the displacements of the Lamb wave was performed using electronic speckle pattern interferometry. From the out-of-plane displacement maps, the wavelength of a single-mode Lamb wave of the known frequency is measured. Numerous Lamb modes at different frequencies are generated, from which the thickness of the plate and two independent elastic constants are determined.

A similar method was used in [10]. The presented paper described the combination of the simplex method and the semi-analytic finite element (SAFE) algorithm to determine elastic constant of a given structure. The propagation of Lamb waves is monitored in order to extract the group velocity dispersion curves in an aluminium plate. A pulsed laser excited the guided wave, and a pair of broadband piezoelectric transducers was used to detect the Lamb wave. The reconstruction of the elastic properties of the plate is performed until the discrepancy between the SAFE and the experimental dispersion curves is minimized. The reconstruction used the A_0_ and S_0_ modes in the frequency range between 100 and 650 kHz.

The hybrid computational system for aluminium plate parameter identification is presented in [11]. The method is based on guided wave measurement and application of artificial neural networks. The Lamb waves were generated by a contact method or laser. The proposed method is non-iterative and using pseudo-experimental data.

Measurement of the thickness of metallic films with an antisymmetric Lamb A_0_ wave mode was discussed in [12]. The Lamb wave was excited by a piezoelectric element and picked up by an electromagnetic acoustic transducer. Therefore, this method is suitable for measurements of metallic films only. It is also necessary to point out that the velocity of the A_0_ mode depends not only on a thickness of the film, but on the Young’s modulus also; however, it was not analyzed in this paper.

In most of the presented papers, the attention is focused on measuring a thickness and elastic properties not of plastic but of aluminium plates. The thicknesses of the plates ranged from 1 to 5 mm, and a laser was used to excite or receive the Lamb waves. Aluminum plates are favorable objects for the study of Lamb waves due to the simple excitation, propagation, and reception of these waves.

The guided Lamb waves were also applied for measurement of elastic properties of thin paper and mineral products [13,14,15]. However, in [13,14] the two-side access air-coupled method is used—the ultrasonic wave is sent through an air gap to the paper sample and is picked up on the other side of the sample. The elastic constant such as the Young’s modulus is found from the comparison of theoretical and experimentally obtained dispersion curves. For investigation of a mineral paper, rather high-frequency ultrasonic waves in the range of 0.15–2.3 MHz were used. Such a method is not convenient for industrial applications due to two-side access approach.

The one-side approach for on-line measurement of the tensile stiffness of the moving anisotropic paper web in a paper machine is presented in [15]. The guided S_0_ wave mode is excited by a dry friction and picked up by air-coupled ultrasonic transducers. The tensile stiffness index is found from the measured velocity of the S_0_ mode. This method was successfully verified in industrial conditions.

Attenuation of a slow A_0_ Lamb wave mode in a polyvinyl chloride (PVC) film was measured by us [16] using cylindrical guided wave propagating inside the film. Application of such wave enabled reduction of diffraction errors and increasing accuracy of attenuation measurements. However, other parameters such as thickness or elastic moduli were not measured.

From the presented review, it follows that almost all publications are devoted to measurement of elastic properties and/or thickness of relatively thick 0.8–3 mm aluminium plates by using contact or semi-contact methods. There are some publications in which are described measurements of composite materials [17,18,19], but there are no publications devoted to measurements of very thin plastic films such as polyvinylchloride (PVC) films using guided Lamb waves. Production of such films only in Europe is 5 million tons annually [20]. We have found that such parameters of PVC films as thickness claimed by a manufacturer do not correspond to the real observed values and significant variations of a thickness across even a small size A4 format samples reach up to 10%. Therefore, development of ultrasonic measurement methods suitable for monitoring quality of the production becomes relevant. For industrial applications, the methods with one-side approach and without direct contact with a film would be preferable because they could be applied for on-line monitoring.

It is necessary to point out that those properties of plastic materials, and especially of PVC films, are very different from metals. The thickness of such films is in the range from 25 µm up to 200 µm instead of thicknesses in the range of a few millimeters characteristic for metallic plates. Propagation velocities of ultrasonic waves including A_0_ and S_0_ guided wave modes in plastic films are significantly lower. For example, the velocity of S_0_ mode in aluminium plates is 5400 m/s and in PVC films it is only 1600 m/s. The attenuation of bulk longitudinal ultrasonic waves in plastics increases with the frequency and is higher than in metallic plates. For example, in plastics such as polypropylene and PVDF, they are between 2 dB/cm at 300 kHz and 5 dB/cm at 500 kHz [21,22]. In the case of guided waves, the frequency dependences of the attenuation are more complicated. Our measurements have shown that, even at the low frequency of 44 kHz, the attenuation coefficient of A_0_ mode in PVC 150 µm thickness film is 2 dB/cm [16]. All of those differences indicate that the methods developed for characterization of metal plates cannot be applied directly for plastic films.

Therefore, the objective of this research was development of the single side access semi-contactless ultrasonic method suitable for simultaneous measurement of the thickness and Young’s modulus of thin plastic films.

The paper is organized as follows. In Section 2 theoretical analysis of propagation of A_0_ and S_0_ guided wave modes in PVC film are analyzed, and a measurement algorithm is proposed. In Section 3, the experimental set-up with which measurements were performed is described and experimental results are presented. In Section 4, conclusions and discussion of the obtained results are given.

## 2. Theoretical Analysis

### 2.1. Properties of the Measured Parameters

Propagation of guided Lamb waves in thin plates and films depends on elastic properties of the material, density, thickness, and frequency of the guided wave. It allows determination of a thickness and some elastic parameters such as Young’s modulus of the structure under a test exploiting the measured phase velocities of different Lamb wave modes.

In most cases, the density of such structures is known or estimated experimentally, the Poisson’s ratio *ν* is often known and in a practice the influence of its variations on propagation velocities is negligible. Therefore, the main parameters characterizing such structures are their thickness *d*, and Young’s modulus *E*.

For investigation of feasibility to measure, simultaneously the thickness and the Young’s modulus we have selected thin PVC film with the thickness in the range up to 250 µm. The properties of the PVC film provided by the manufacturer (United States Plastics Corporation VYCOM, Scranton, PA, USA) are presented in Table 1 [20].

First, we have calculated dispersion curves of guided wave modes by the semi-analytic finite element method (SAFE) [5]. Calculation by this method in the case of a thin film or plate with infinite lateral dimensions of the plate is based on splitting it into a finite number of *M* thin layers, each of which is described in one axis direction. In the second axis direction, it is assumed that the plate is infinite. Therefore, the particle displacement of any point in an element is given by
(1)$$u^{\left(e\right)}=N\left(\zeta \right)U^{\left(e\right)}e^{i\left(kx-\omega t\right)}$$
where **N**(*ζ*) is the matrix of the shape function, *ζ* is the variable in the local coordinate system, **U**^(*e*)^ are the nodal displacements of the element, *x*-axis is the wave propagation direction, *k* is the wavenumber, *ω* is the angular frequency, and *t* is the time.

After a standard finite element, assembling procedure the following linear system of algebraic equations in the global coordinate system is obtained
(2)$$\left(K_{1}+ikK_{2}+k^{2}K_{3}-\omega^{2}M\right)U=0$$
where the **U** represents the global vector of nodal displacements, **K**_1_, **K**_2_, **K**_3_, and **M** are the matrices in the global coordinate system. The solutions of this equation describe the propagation characteristics of the Lamb wave in a film and the results are presented as dispersion curves.

Analysis of the calculated dispersion curves showed that in the 150 µm thickness film at the frequencies lower than 50 MHz only two Lamb wave modes A_0_ and S_0_ may propagate. The dispersion curves of those modes in the frequency range 0–300 kHz are shown in Figure 1.

From the results presented in Figure 1a, it follows that the phase velocity of the A_0_ mode strongly depends both on a thickness *d* of the film and the frequency of the ultrasonic signal *f*. Contrarily, the phase velocity of the S_0_ mode in the low frequency range (<300 kHz) almost does not depend either on the frequency or on the thickness of the film. This conclusion is valid for films of different thickness. Taking that into account, it is possible to assume that the velocity of the S_0_ mode does not depend on the thickness of the film, but it should depend on elastic properties of the film, particularly on the Young’s modulus. Correspondingly, the velocity of the A_0_ mode should depend on both parameters—the thickness and the Young’s modulus.

It is necessary to point out that the A_0_ mode phase velocity’s sensitivity to variations of the *f·d* product is biggest in the range of low *f·d* values—e.g., less than 20. In the case of the 150 µm thickness film, it corresponds to the frequencies lower than 100 kHz (Figure 1b). Hence, for precise thickness measurements, the frequency of the A_0_ mode should be also selected lower than 100 kHz. It means that the thickness and the Young’s modulus of the film can be found exploiting the measured phase velocities of the S_0_ and A_0_ modes.

For excitation of A_0_ and S_0_ modes, different frequencies were selected. For the S0 mode excitation the fixed frequency *f =* 180 kHz was selected because the dispersion curve of this mode in the low frequency range (<300 kHz) is independent of the frequency and the same measurement results would be obtained at other frequencies. For excitation of the A_0_ mode, the lower frequency *f =* 50 kHz was chosen because according to the modeling results at this frequency the best sensitivity to variations of the film thickness is obtained [23].

To check those conclusions, we have performed calculations of the phase velocities of both modes versus the Young’s modulus and the thickness of the film. The results are presented in Figure 2a,b.

The calculations were performed at the fixed frequency 50 kHz. In Figure 2a, dependencies of the phase velocities of the A_0_ and S_0_ modes for the fixed film thickness *d* = 150 µm and for the fixed Young modulus *E* = 2.937 GPa (Figure 2b) are presented. The obtained results show that as expected the phase velocity of the S_0_ mode strongly depends on the Young’s modulus, but almost is not affected by the thickness *d* of the film. The phase velocity of A_0_ mode depends on both parameters, but the thickness of the film influences the phase velocity much more than the phase velocity of the S_0_ mode.

Based on those conclusions, it is possible to assume that, by knowing the dispersion curves of the A_0_ and S_0_ modes in a certain frequency range, it is possible to estimate the thickness and the Young’s modulus of the film under study.

### 2.2. Measurement Algorithm

For estimation of a film thickness and the Young’s modulus using velocity measurements, the relations of those parameters to the phase velocities of A_0_ and S_0_ modes are necessary. It is not possible to get them in an analytic way, but they can be found numerically from the set of dispersion curves calculated for films of different thickness and different Young’s modulus. The relations between the film thickness *d*, the Young’s modulus *E* and the phase velocities of both modes obtained numerically were approximated by third order polynomials:*d*_A_ = 3.95 × 10^−6^·*c*_phA_^3^ + 0.006·*c*_phA_^2^ + 0.075·*c*_phA_−2.44 (µm),(3)
*d*_S_ = −2.82 × 10^5^·*c*_phS_^3^ + 1.35 × 10^9^·*c*_phS_^2^−2.16 × 10^12^·*c*_phS_ + 1.15 × 10^15^ (µm), (4)
*E*_A_ = 4.19 × 10^−6^·*c*_phA_^3^−8.63 × 10^−4^·*c*_phA_^2^ + 0.08·*c*_phA_−2.75 (GPa),(5)
*E*_S_ = −3.97 × 10^−18^·*c*_phS_^3^−1.15 × 10^−6^·*c*_phS_^2^−4.28 × 10^−11^·*c*_phS_ + 9.21 × 10^−5^·(GPa), (6)
where *c*_phA_ is the phase velocity of the A_0_ mode and *c*_phS_ is the phase velocity of the S_0_ mode. Both velocities in those approximations are in m/s. The subscripts A and S indicate that the parameter is related to A_0_ or S_0_ mode.

From the performed calculations, it follows that the phase velocity of the S_0_ mode does not noticeably depend on the thickness *d* of the film; therefore, it should be exploited only for determination of the Young’s modulus.

The obtained results allow formulating the following algorithm for estimation of the Young’s modulus *E* and the film thickness *d*:The phase velocity *c*_phS_ of the S_0_ mode at the excitation frequency *f* is measured.From this velocity, the Young’s modulus *E*_S_ is found from the determined approximation:
*E*_S_ = A·*c*_phS_^3^ + B·*c*_phS_^2^ + C·*c*_phS_ − D,(7)
where A, B, C, and D are the determined approximation coefficients.The phase velocity *c*_phA_ of the A_0_ mode at the excitation frequency *f* is measured.The thickness of the film can be found from the measured phase velocity *c*_phA_ of the A_0_ mode in two different ways. In the first case for that, the approximation given in Equation (3) is exploited. In the second case, the thickness of the film is determined using the expression
(8)$$d=\frac{c_{\mathrm{phA}}^{2}}{2\pi f}\sqrt{\frac{12\rho \left(1-\nu^{2}\right)}{E_{S}}}$$
where *ρ* is the density of the film, which is measured beforehand, and *E*_S_ is the Young’s modulus value obtained from Equation (7).

Equation (8) is derived from the well-known formula proposed by Cremer using the resonance method [24]
(9)$$c_{\mathrm{phA}}^{2}=2\pi fd\sqrt{\frac{E}{12\rho \left(1-\nu^{2}\right)}}$$

This formula is valid for homogeneous thin plates in a low frequency range, which in our case is up to 100 kHz.

For thickness measurements, Equations (3) and (8) were proposed. Equation (8) presented in the described measurement algorithm is preferable because it allows to calculate the film thickness faster and easier. Using Equation (3), new approximation curves should be calculated from the obtained experimental results and only then the film thickness could be estimated.

The proposed algorithm consists of several previously investigated and verified algorithms, which are described in detail and investigated in or publications [24,25,26]. The new algorithm is designed as a combination of those algorithms for solving newly raised problems.

### 2.3. Guided Wave Excitation and Reception Method

For performing measurements of velocities of A_0_ and S_0_ modes the excitation and reception methods of those modes should be selected. For practical applications, for example for on-line measurements during manufacturing process, non-contact, e.g., air-coupled methods should be more preferable.

The excitation method very much depends on selection of the frequency *f* of guided waves to be excited (Figure 3).

In the case of A_0_ mode, there are two different frequency ranges. The first one is the low frequency range in which velocity of the A_0_ mode is lower than the ultrasound velocity in air *c*_air_ = 342 m/s. In this frequency region, the A_0_ mode does not excite a leaky wave in surrounding air. The second one is the range of higher frequencies in which the velocity of this mode is higher than the ultrasound velocity in air and its propagation is accompanied by a leaky wave in air. Excitation of guided waves in those two different frequency ranges by air-coupled ultrasonic transducers is different.

For excitation and reception of ultra-slow A_0_ mode, very efficient lead magnesium niobate-lead titanate (PbMg1/3Nb2/3O3-PbTiO3) PMN-32%PT air-coupled transducers and arrays can be used [18]. For the film of 150 μm thickness, the phase velocity of the A_0_ mode is slower than the ultrasound velocity in air in the frequency range up to 300 kHz (Figure 3). The highest frequency at which *c*_phA_ becomes slower than *c*_air_ is increasing when the thickness of the film is decreasing. The lower frequency range in the case of thin films is more attractive because, in it, the sensitivity of the phase velocity to thickness variations is higher than at higher frequencies. Consequently, in this frequency range, it is possible to obtain a higher accuracy of thickness measurements.

At higher frequencies air-coupled transducers deflected with respect to the normal to the surface of the film at the angle *α*_opt_ given by the Snell’s law can be exploited
(10)$$\alpha_{\mathrm{opt}}=\sin^{-1}\frac{\lambda_{\mathrm{air}}}{\lambda_{A_{0}}\left(f\right)}=\sin^{-1}\frac{c_{\mathrm{air}}}{c_{A_{0}}\left(f\right)}$$
where *λ*_air_ and *λ*_A0_(*f*) are wavelengths correspondingly in air and in the film.

The phase velocity of the S_0_ mode *c*_S0_(*f*) always is higher than the ultrasound velocity in air *c*_air_; therefore, it can be excited by an air-coupled transducer deflected according to Snell’s law.

The optimal deflection angles of air-coupled ultrasonic transducers versus the frequency for films of two different thicknesses are shown in Figure 4.

From the presented results, is follows that—in the low frequency range up to 230 kHz in the film of 200 μm thickness—to excite the A_0_ mode by a deflected air-coupled transducer is impossible because the required deflection angle with respect to the normal to the film surfaces becomes 90°, which is impossible to realize. The same conclusion is valid for the film of 150 μm in the frequency range up to 305 kHz. Therefore, for excitation of the A_0_ mode in this frequency range, PMN-32%PT arrays can be exploited [18].

Efficiency of air-coupled excitation of S_0_ mode is lower than of the A_0_ mode, especially in the lower frequency range, due to much higher velocity of this mode, the wavelength is much longer and consequently the ratio of the excitation area diameter to the wavelength is much smaller. In such a case, the efficiency of excitation reduces. Therefore, for excitation of the S_0_ mode, we have selected the frequency range higher than 100 kHz. For this, we propose to use air-coupled flat ultrasonic transducer deflected from the normal to the surface of the film at the angle given by Snell’s formula (Equation (10)).

## 3. Experimental Verification

### 3.1. Experimental Set-Up

For verification of the proposed measurement algorithm, we performed experimental studies of two polyvinyl chloride (PVC) films of different thickness (*d*_1_ = 150 µm and *d*_2_ = 200 µm) but with identical elastic constants according to the manufacturer (Table 1). The film with lateral dimensions 210 × 297 mm^2^ was fixed in a PVC film-mounting bracket (Figure 5).

During experiments, the phase velocities of S_0_ and A_0_ Lamb wave modes were measured. Considering that S_0_ mode is rather difficult to excite by the air-coupled method, the contact excitation and reception was used. For that contact point type transducers with a hemispherical plastic tip and with 180 kHz resonant frequency were exploited. Experiments were performed using a scanner Standa 8MTF-75LS05 (Standa Ltd., Vilnius, Lithuania) and the ultrasonic measurement system “Ultralab” [27] (Figure 6). All units including the ultrasonic transducers and the ultrasonic system “Ultralab” were developed at Ultrasound Research Institute of the Kaunas University of Technology. The ultrasonic transducers possessed small diameter plastic tips that allowed the performance of dry contact B-scans which did not damage the PVC film.

The transmitter was excited by the 500 V, 180 kHz, three-period sinusoidal signal with the Gaussian envelope. The ultrasonic receiving transducer was scanned with respect to the ultrasonic transmitter in the range of 57–87 mm with 0.1 mm steps.

The obtained B-scans of the measured amplitudes of the Lamb wave signals in the PVC films of different thickness are shown in Figure 7. The amplitudes of the received signals normalized with respect to the maximal value are shown color coded.

In the collected B-scans, the fastest S_0_ mode is clearly visible and is suitable for a phase velocity measurement.

The A_0_ mode was excited by a contactless air-coupled method and registered by Polytec 500 laser interferometer (Polytec GmbH, Waldbronn, Germany) (Figure 8).

The A0 mode was generated by the single rectangular strip-like PMN-32%PT piezoelectric element (HC Materials Corporation, Bolingbrook, IL, USA) with dimensions of 15 × 5 × 1 mm^3^ in which the transverse-extension mode was excited [5]. The piezoelectric element was excited by the 6 V amplitude 50 kHz center frequency three-period pulses. The ultrasonic wave was radiated through the 1 mm air gap perpendicular to the film surface. For improvement of the sensitivity and widening of the transducer bandwidth, a quarter-wavelength matching layer on the tip of the piezoelectric strip was used.

The normal displacement waveforms of the film were registered by the Polytec laser interferometer. To improve the signal-to-noise ratio at each measurement point, eight signals were averaged. The film was scanned in the *x*-axis direction when the distance from the transducer was *x*_1_ = 20 mm and the probing distance was *x*_2_ = 40 mm, the scanning step was d*x* = 0.1 mm. Figure 9a shows the obtained B-scan image by a non-contact scanning of the *d*_1_ = 150 µm thick PVC film. The observed amplitude spectrum of the relatively short ultrasonic pulse due to influence of the quarter-wavelength matching layer possesses two peaks, and one of them is at the frequency 52 kHz (Figure 9c).

As the picked-up ultrasonic signals are distributed in a relatively narrow frequency range (Figure 9a,c), it was decided to filter the obtained B-scan image with a narrow band Gaussian filter. The frequency response of the filter was close to the shape of the informative signal with the resonant frequency of 50 kHz and the bandwidth of 5 kHz. The filtered B-scan image is shown in Figure 10a.

The obtained S_0_ and A_0_ mode signals were used for the phase velocity measurements presented in the next chapter.

### 3.2. Measurement Results

At the beginning of the experimental studies, the parameters declared by the PVC film manufacturers presented in Table 1 were verified. The thickness of the films was measured with the micrometer (ATORN, Hommel Hercules, Germany, measurements error ±5 µm). Measurements of the thickness and density of the films showed that the thicknesses of the *d*_1_ = 150 µm and *d*_2_ = 200 µm films were correspondingly *d*_1_ = 135 ± 5 µm and *d*_2_ = 180 ± 5 µm, e.g., rather different than those claimed by the manufacturer. The density of those films was *ρ* = 1340 kg/m^3^. Those parameters were further used in calculations of the Lamb wave A_0_ and S_0_ modes phase velocity dispersion curves.

The Lamb wave S_0_ mode phase velocity dispersion curves for both films were reconstructed from the B-scan images shown in Figure 7. The 2D-FFT spectra of those B-scans (color) and the reconstructed phase velocities of the S_0_ mode (dots) are shown in Figure 11a,b. The method used for evaluation and reconstruction of the segments of those curves exploiting only two signals measured at two close points was described in detail in [25]. The metrological performance of this technique was described in [26].

The phase velocities *c*_phS_ of the S_0_ mode in the films of different thickness at the excitation frequency *f* = 180 kHz obtained by the proposed reconstruction algorithm, were correspondingly *c*_phS_ = 1821 ± 35 m/s for the film *d*_1_ = 135 µm and *c*_phS_ = 1859 ± 37 m/s for the film *d*_2_ = 180 µm. The difference between the S_0_ mode velocities in films of different thickness is only 38 m/s or 2%—e.g., it almost does not depend on the thickness, but strongly depends on the Young’s modulus *E*.

The dependences between the Young’s modulus *E*_S_ and the measured phase velocity c_phS_ were obtained from the experimentally reconstructed sections of the dispersion curves (Figure 11) and approximating them by the third-order polynomials. Those dependences for films of different thickness are different and must be determined in advance. In our case, they are given by the following polynomials:*E*_S_ = −6.91 × 10^−16^·*c*_phS_^3^ + 1.1 × 10^−6^·*c*_phS_^2^ − 9.06 × 10^−9^·*c*_phS_ + 1.1 × 10^−3^·(GPa) (*d*_1_ = 135 µm),(11)
*E*_S_ = −2.19 × 10^−16^·*c*_phS_^3^ + 1.1 × 10^−6^·*c*_phS_^2^ − 2.87 × 10^−8^·*c*_phS_ + 2 × 10^−3^·(GPa) (*d*_2_ = 180 µm).(12)

The values of the Young’s modulus determined from those approximation curves are *E*_S_ = 3.66 GPa for the film *d*_1_ = 135 µm and *E*_S_ = 3.82 GPa for the film *d*_2_ = 180 µm.

It is not necessary to determine film coefficients of a certain thickness in advance. It is enough to know the approximate limits of the Young’s modulus and the film thickness which can be preliminarily measured mechanically.

The phase velocities *c*_phA_ of the A_0_ mode were found from the filtered B-scan (Figure 10a) at the excitation frequency *f* = 50 kHz using the proposed algorithm. Their values are *c*_phA_ = 147.8 ± 2.9 m/s for the film *d*_1_ = 135 µm and *c*_phA_ = 170.8 ± 3.4 m/s for the film *d*_2_ = 180 µm.

Using the values of the experimentally determined Young’s modulus *E*_S_ and the A_0_ mode velocity *c*_phA_ the thickness *d* of the PVC film was calculated according to Equation (8). The calculation results are presented in Table 2.

### 3.3. Accuracy of Measurements

Uncertainty of the thickness and the Young’s modulus measurements first depends on an accuracy of phase velocity measurements. This uncertainty can be evaluated from the approximated relations of those parameters and the corresponding phase velocities (Equations (3)–(6)). It is possible to evaluate how much variation of the phase velocities caused by measurement errors affect the potential accuracy of estimation of the thickness *d* and the Young’s modulus *E* of the film under a study. In Figure 12a,b, variations of those parameters when the phase velocity *c*_ph_ is estimated with a measurement error up to Δ*c*_ph_ = ±1% are presented.

From the presented calculation results, it follows that the uncertainty of the A_0_ mode phase velocity Δ*c*_ph_ = ±1% leads to the uncertainty of the film thickness Δ*d*_A_ = ±2% (Figure 12b). In our case, when the absolute thickness of the film is 150 μm, it is Δ*d*_A_ = ±3 μm. Correspondingly, the uncertainty of the S_0_ mode phase velocity Δ*c*_ph_ = ±1% leads to the uncertainty of the Young’s modulus Δ*E*_S_ = ±2% (Figure 12a).

The elastic parameters and correspondingly the ultrasound velocity in PVC material depend on a temperature. It was found that the velocity of longitudinal ultrasonic waves decreases linearly versus temperature in the temperature range of 20 to 40 °C. The observed decrease is −3.143 ± 0.076 m/s per 1 °C, which is 0.2% [28].

Variations of the temperature by 1 °C lead to the uncertainty of the film Young’s modulus measurement 0.8% for A_0_ mode and 0.4% for S_0_ mode respectively. The temperature change by 1 °C leads to an uncertainty of the film thickness measurement of 0.4% using the A_0_ mode.

For measurements, we used short wide band ultrasonic pulses and the frequency 50 kHz corresponds to the central frequency of the electric pulse used for excitation of the ultrasonic transducer. In the PVC film, an ultrasonic pulse propagates, the spectrum of which is shown in Figure 9b. Due to the frequency response of the ultrasonic transducer with the matching layer, the maximum is observed at 52 kHz. Therefore, for thickness measurements the frequency 52 kHz was used. One of the uncertainty components of the A_0_ mode phase velocity measurements may be due to the frequency deviation. However, the influence of the frequency deviation on the thickness measurement is by one order smaller than the influence of the measurement uncertainty of the S_0_ mode phase velocity. Change of the frequency by 2 kHz (from 50 to 52 kHz) causes only a 0.1% change in the film thickness measurement for a 200 μm thick film (Equation (8)) and only 0.07% for a 150 μm thick film.

The accuracy of the performed thickness measurements was evaluated by the measurement uncertainty. The expanded relative uncertainty *δ_d_* of the film thickness *d* was obtained by
(13)$$\delta_{d}=100\%\cdot \frac{\left|d_{\mathrm{meas}}-d_{\mathrm{ultr}}\right|}{d_{\mathrm{meas}}}$$
where *d*_meas_ is the thickness measured by the micrometer and *d*_ultr_ is the thickness measured by the proposed method.

The film thickness first was measured with a micrometer by the entire perimeter of the film under test with 20 mm steps. Overall, 50 measurements were performed. The measuring accuracy of the micrometer was ±5 µm. The expanded relative uncertainties of thickness measurements are given in Table 2. Table 2 shows the average values of the measurements performed. The uncertainty due to the measured 200 µm film value is approximately twice that of the 150 µm film due to the higher slope of the A_0_ mode dispersion curve for the thicker film in the same frequency range (Figure 3).

## 4. Discussion and Conclusions

The aim of this work was to develop and to investigate a one-side access ultrasonic method suitable for simultaneous measurement of the thickness and the Young’s modulus of thin plastic films. To achieve this goal, the excitation, propagation, and reception of the Lamb waves in thin PVC films were studied. It was found that the methods used for characterizing thin metal plates cannot be directly applied to thin plastic films. This is due to the much lower velocities of the guided Lamb waves propagating in plastic films and a higher attenuation of those waves. In the case of thin plastic films, the phase velocity of the A_0_ mode exploited for thickness measurements becomes slower than the ultrasound velocity in air and there is no leaky wave radiated by the film into the air. In the case of metallic films, in most cases, phase velocity of A_0_ mode is higher than in surrounding air and its propagation is accompanied by a leaky wave. That requires different excitation and reception methods of ultrasonic guided Lamb waves.

It was found that the phase velocity of the symmetrical S_0_ mode does not depend on the thickness of the film but depends on the Young’s modulus. Contrarily, the phase velocity of the A_0_ mode depends both on the thickness and on the Young’s modulus of the film. Exploiting measurements of phase velocities of those two modes, the measurement algorithm for evaluation of the thickness and the Young’s modulus was proposed. It was found that the highest sensitivity of the A_0_ mode to variations of the film thickness is observed in the low frequency range up to 100 kHz, in which the phase velocity of this mode is lower than the ultrasound velocity in air. It means that, in this frequency range, it is impossible to excite the A_0_ mode by conventional means using deflected air-coupled transducers. In order to overcome this problem, we have proposed using high efficiency PMN-PT piezoelectric crystals, which allows exciting the A_0_ mode without contact. To excite S_0_ mode in a contactless way is problematic, therefore for this purpose we have used contact point type ultrasonic transducers with a hemispherical plastic tip, which provided a reliable dry acoustic contact with a film and left no traces on the film surface when it was moving. In principle, it is possible to use a multi-element phased array with PMN-PT piezoelectric elements for excitation of the S_0_ mode in a low frequency range [29].

The performed measurements demonstrated feasibility to simultaneously measure the thickness and the Young’s modulus of thin PVC films by means of guided ultrasonic waves. It is necessary to point out that the measured thickness of PVC films significantly differed from the values declared by the manufacturer. The method developed looks promising for a contactless on-line monitoring of the mentioned parameters. In this case, for contactless reception of guided waves, the air-coupled array and corresponding signal processing procedure developed by us and described in [30] could be used.

## Figures and Tables

**Figure 1 sensors-21-06779-f001:**
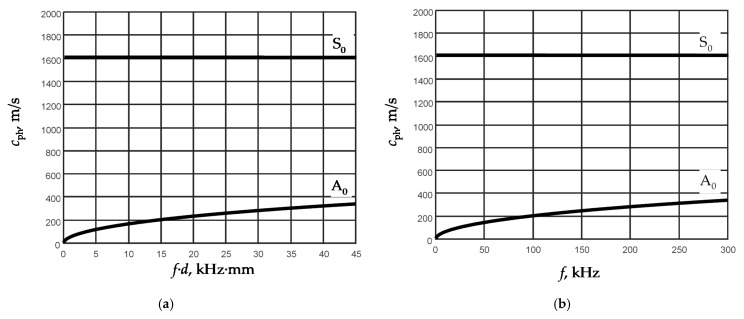
Phase velocity dispersion curves of Lamb wave fundamental modes calculated by SAFE method: (**a**) versus *f*·*d*, kHz·mm; (**b**) versus frequency *f* in *d* = 150 µm thickness PVC film.

**Figure 2 sensors-21-06779-f002:**
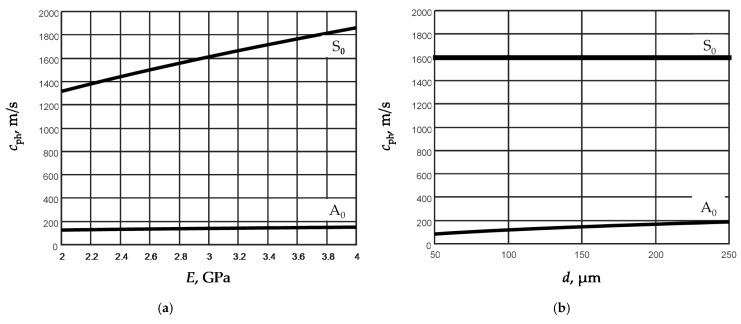
Dependences of the phase velocities of Lamb wave modes A_0_ and S_0_ in the PVC film on the Young’s modulus *E* (**a**) and the film thickness *d* (**b**) when the film thickness *d* = 150 µm (**a**) and the Young’s modulus *E* = 2.937 GPa (**b**).

**Figure 3 sensors-21-06779-f003:**
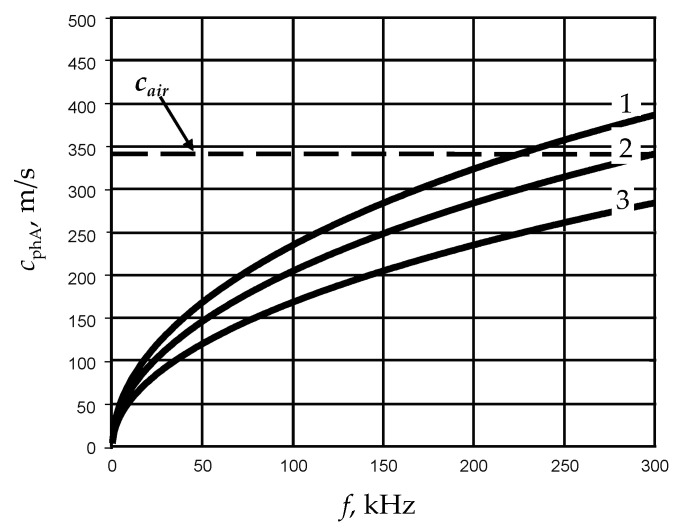
Phase velocities of A_0_ mode in PVC films of different thickness: 1—*d* = 200 μm, 2—*d* = 150 μm, 3—*d* = 100 μm.

**Figure 4 sensors-21-06779-f004:**
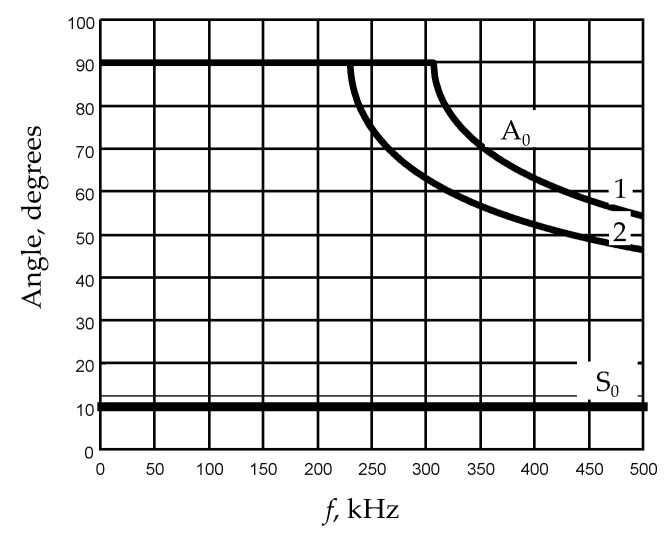
Dependences of the deflection angles of air-coupled ultrasonic transducers for PVC films of different thickness: 1—*d* = 150 μm, 2—*d* = 200 μm.

**Figure 5 sensors-21-06779-f005:**
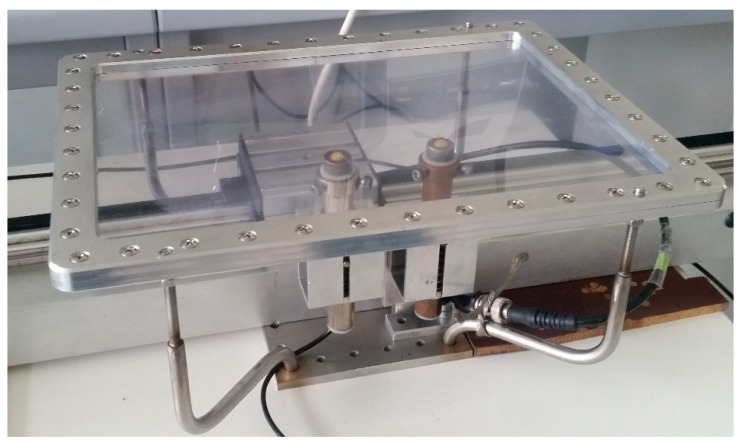
View of PVC film mounting bracket and ultrasonic transducers below the film.

**Figure 6 sensors-21-06779-f006:**
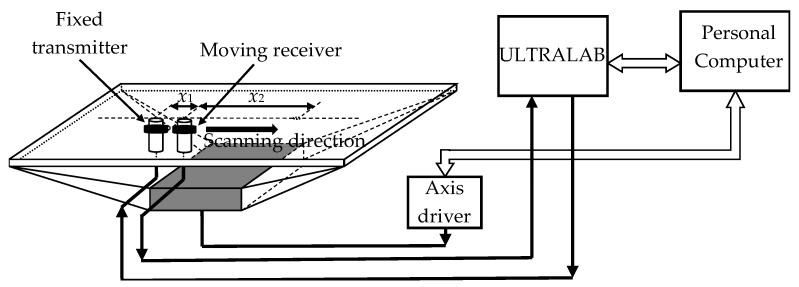
Structural scheme of the Lamb wave S_0_ mode signal generation and recording in thin PVC film.

**Figure 7 sensors-21-06779-f007:**
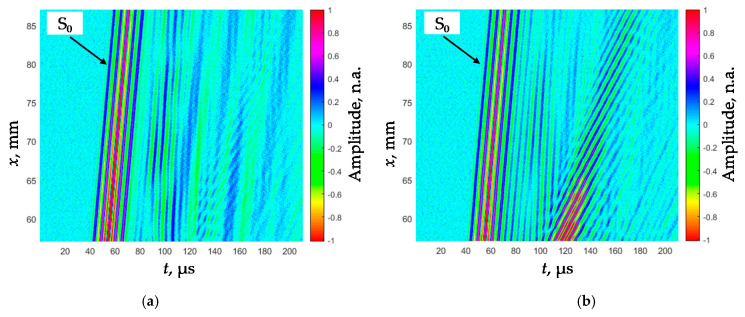
B-scans of the measured amplitudes of the Lamb wave signals in the PVC films of different thickness: (**a**) *d*_1_ = 150 µm, (**b**) *d*_2_ = 200 µm.

**Figure 8 sensors-21-06779-f008:**
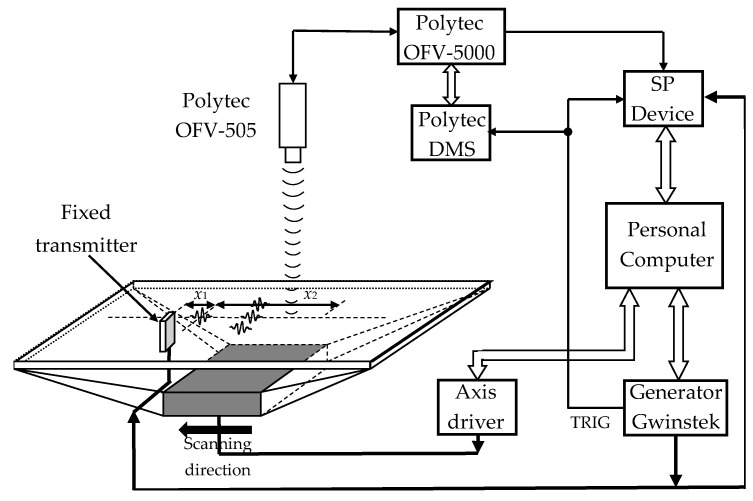
Structural scheme of the Lamb wave A_0_ mode signal generation and recording in thin PVC film.

**Figure 9 sensors-21-06779-f009:**
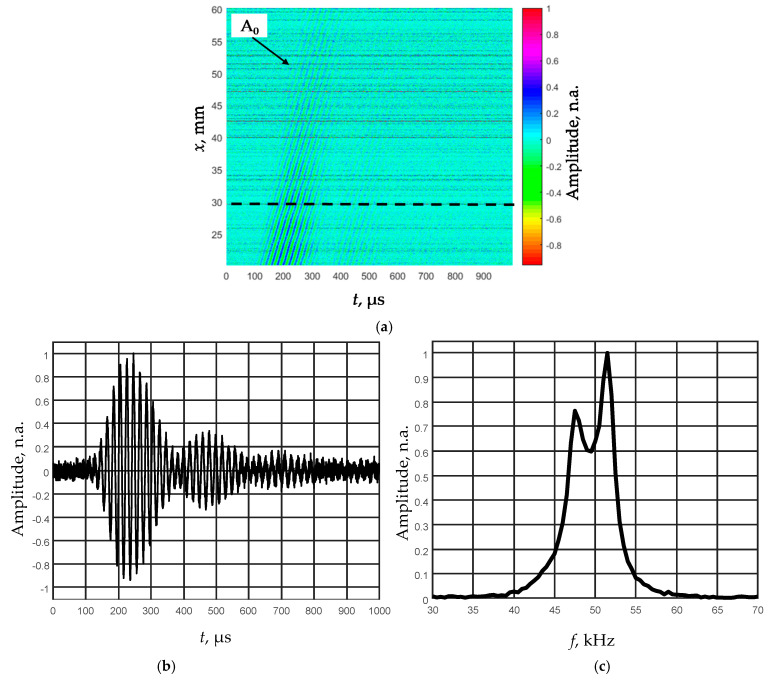
Air-coupled excitation of the A_0_ mode at the low frequency (50 kHz) in 150 µm thickness PVC film: the measured B-scan image of the normal displacements (**a**), the recorded signal at the distance 30 mm from the transmitter (**b**), and the spectrum of this signal (**c**).

**Figure 10 sensors-21-06779-f010:**
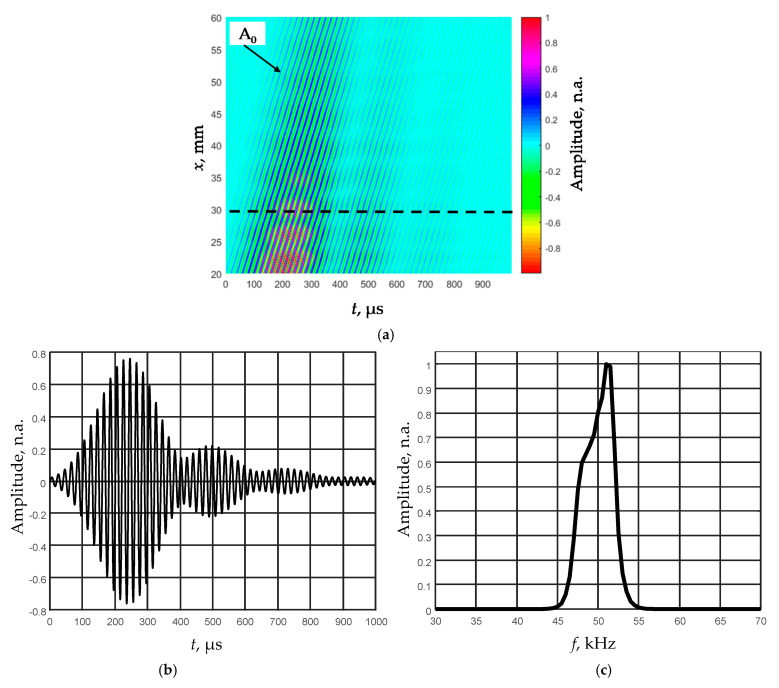
Air-coupled excitation of the A_0_ mode at the low frequency (50 kHz) in 150 µm thickness PVC film: B-scan image of the normal displacements obtained after filtering of the ultrasonic signals with the Gaussian filter (**a**), the filtered signal at the distance of 30 mm from the transmitter (**b**), and the frequency response of this signal (**c**).

**Figure 11 sensors-21-06779-f011:**
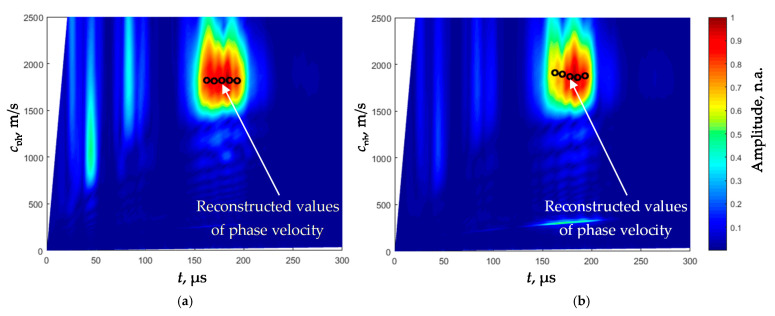
2D-FFT spectra of the B-scans of the recorded Lamb wave signals (color) and the reconstructed phase velocities of the S_0_ mode using the proposed algorithm (dots) of the PVC films with different thickness: (**a**) *d*_1_ = 135 µm, (**b**) *d*_2_ = 180 µm.

**Figure 12 sensors-21-06779-f012:**
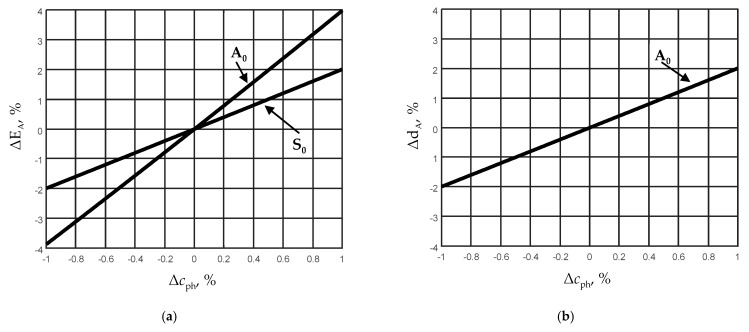
Measurement uncertainties of the Young’s modulus *E* (**a**) and film thickness *d* (**b**) versus phase velocity *c*_ph_ measurement errors of A_0_ and S_0_ modes.

**Table 1 sensors-21-06779-t001:** Parameters of the PVC film (Vintec^®^ Clear PVC).

Parameter	*E*, GPa	*ν*	*ρ**,* kg/m^3^
PVC film	2.937	0.42	1400

**Table 2 sensors-21-06779-t002:** Calculation results of the PVC film thickness.

PVC Film Thickness Declared by the Manufacturer, *d*, µm	Film Thickness Measured by the Micrometer, *d*_meas_, µm	Film Thickness Measured by the Proposed Ultrasonic Method *d*_ultr_, µm	Expanded Relative Uncertainty δ*_d_*,%
150	135 ± 5	132 ± 2.7	2.07
200	180 ± 5	173 ± 6.7	3.89

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
