# Peer review of "Simultaneous Measurement of Thickness and Elastic Properties of Thin Plastic Films by Means of Ultrasonic Guided Waves"

_sensors, 2021, doi:10.3390/s21206779_

Round 1
Reviewer 1 Report
The simultaneous measurement of the thickness and the Young’s modulus of was thin plastic films was investigated. It is an interesting topic. However, many points need to be clarified.
- How Equations 11 and 12 were obtained?
- For plastic films of different thicknesses, the coefficients are completely different. If it is necessary to determine the coefficients for a certain thickness film in advance?
- From Figure 1, the phase velocity of the S0 mode almost does not depend on the thickness of the film. Is the conclusion suitable for all thickness?
- Temperature is an important factor affecting the elastic property and the propagation velocity. In this paper, there is no experiment or discussion on the effect of the temperature.
- Some labels in the figures are hided.
Author Response
We would like to thank Reviewer 1 for comments and proposals how to improve the submitted manuscript. We have made the necessary corrections and added a novel text where it was necessary.

Reviewer 2 Report
Title: Simultaneous measurement of thickness and elastic properties of thin plastic films by means of ultrasonic guided waves
Summary:
The authors demonstrated experimentally using non-contact guided waves for thickness measurement technique of thin PVC film. From simulations, authors noted that Young's modulus and thickness correlated independently to S0 and A0 waves respectively.
Comments:
- All the images are in very low resolution, authors will need to put some more effort before next submission.
- Reviewer understand that the back-calculation algorithm is derived based on simulation results to generate dispersion curves. Have the authors validate the model before relying too much on the it?
- Have the authors verified the proposed algorithm with any simulated scenarios?
- Will temperature affect the performance of the proposed algorithm since PVC's properties is highly depending on temperature.
- Reviewer is aware that the air-coupled transducers have limitation on the range of sensing (not suitable for high frequency). Why the authors only perform experiment using a single frequency excitation? How about other frequency? More frequencies will show a better understanding of the proposed algorithm.
- 2% to 9.44% error for such a thin material for a measuring technique is worrying the reviewer. How can the authors improve on the accuracy?
- Micrometer is not a common method to measure the thickness of the film. What is the accuracy of the technique? How many measurement did the authors did to get what have been presented in Table 2?
- How reliable is the proposed technique? Is it repeatable? Authors did not present any replication of testing. One test for each of the two samples may not sound enough to be publish in a journal.
Author Response
We would like to thank Reviewer 2 for comments and proposals how to improve the submitted manuscript. We have made the necessary corrections and added a novel text where it was necessary.

Reviewer 3 Report
This paper describes the measurement method for thickness and Young's modulus of thin plastic films using Lamb waves. Moreover, the experimental results for PVC films with thickness of 150 and 200 µm are shown to verify the method.
I think that the following points should be published before the publication.
The accuracy of the Young's modulus measurement should be also investigated. If the thickness and the Young's modulus measurements mainly depend on the phase velocities of Lamb waves, the experimental errors of the phase velocities should be investigated.
- How much are the errors?
- What are the causes of the errors?
- The center frequency [about 52 kHz in Fig. 9(b) or 10(b)] of A0 mode is different from the input frequency (50 kHz). Is there any effect?
- Why is the error of the thickness for 200 µm film larger than that for 150 µm film?
- Are 4-digit significant figures of the phase velocities valid?
Two thickness measurements using each of Eqs. (3) and (8) are described on page 13, lines 441-442, but the method using Eq. (3) is not used. Please explain the reason.
It is described on page 13, lines 445-447 that the uncertainties of the thickness and the Young's modulus measurements can be evaluated from Eqs. (3)-(6). However, should the uncertainly of the thickness measurement be evaluated from Eq. (8) rather than Eqs. (3) and (4). Similarly, should the uncertainly of the Young's modulus measurement be evaluated from Eqs. (11) and (12) rather than Eqs. (5) and (6)?
Abstract, "The uncertainty of thickness measurements of the thinner 150 µm PVC film is 2%", I think that the uncertainty of 200 µm film measurement should be also described (to avoid overestimation).
Figs. 1-4, 9(b), 9(c), 10(a), 10(c), and 12: The positions of the vertical scale values are slightly shifted downward.
Figs. 1(b), 2, 3, 11, and 12(b): The vertical labels cannot be seen.
Figs. 7, 9, and 10: The figures (particularly, the scale value) are unclear.
"PVC" and "PMN-32%PT": It should be specified what the abbreviation is (at the first place).
Page 8, line 300, “0÷305 kHz": Is it a clerical error?
Page 9, line 345, "the ultrasonic system "Ultralab": An overview of the system should be given, or a reference about the system should be cited.
Author Response

(The authors gave the same response as above.)

Round 2
Reviewer 1 Report
Most my questions have been addressed. From the results presented in Figure 1, conclusion that the phase velocity of the S0 mode almost does not depend either on the frequency or on the thickness of the film cannot be drawn. More simulation results or references are needed to support this argument.
Author Response
Most my questions have been addressed. From the results presented in Figure 1, conclusion that the phase velocity of the S0 mode almost does not depend either on the frequency or on the thickness of the film cannot be drawn. More simulation results or references are needed to support this argument.
Figure 1,a shows the dependence of the phase velocity on the frequency multiplied by the thickness of the material. At low frequencies (in our paper <300kHz), the phase velocity of the S0 mode does not depend on the frequency and the thickness. This has been confirmed by both theoretical and experimental studies performed by other researchers [5].
Reviewer 2 Report
Thank you authors for addressing the comments.
- Can the authors provide the brand and type of micrometer used? It is weird that the current type of micrometer has +- 5micron error. According to industry standards, it should either be +-2 micron (mechanical) and +-1 micron for digital.
- The temperature information as mentioned in Section 3.3. is not back up with any experimental or simulation data. Can the authors provide more information? How long does each experiment take? Is the experiment conducted in a temperature-controlled environment?
Author Response
1. Can the authors provide the brand and type of micrometer used? It is weird that the current type of micrometer has +- 5micron error. According to industry standards, it should either be +-2 micron (mechanical) and +-1 micron for digital.
The thickness of the films was measured with the micrometre ATORN 30370001 DIN 8631, Hommel Hercules, (Germany). The manufacturer gives the measurements error ±5 microns.
This explanation was inserted in the manuscript.
2. The temperature information as mentioned in Section 3.3. is not back up with any experimental or simulation data. Can the authors provide more information? How long does each experiment take? Is the experiment conducted in a temperature-controlled environment?
Temperature influence on the phase velocity in PVC material was obtained experimentally by other researchers [28] and these data were used by us evaluate the measurement uncertainty due to temperature variations. Our experiments were performed in a laboratory that automatically maintains a standard room temperature (22±0.5oC). Each experiment lasts 0.5 hour.
Reviewer 3 Report
I recommend further revisions for the following points.
I think that the error "in the experiments in this study" [not (only) the theoretical and analytical error but the experimental error] should be comprehensively investigated. Then, it is better to show concretely what error occurred and how much. Moreover, it is better to clarify the following points.
- A0 mode velocity depends on the frequency, and therefore, if the output frequency is different from the input frequency or if the frequency bandwidth exists, will there be an error or increased uncertainty?
- Why is the uncertainty about the measured value of the 200 µm film about twice as large as that of the 150 µm film? (Is it possible to less accurately measure in principle when the film is thicker?)
- Why are both measured values of 150 and 200µm films smaller than the measured value using a micrometer?
Given that the uncertainty is a few percent, isn't three significant digits reasonable?
For "PVC" in the abstract, it should be specified what the abbreviation is.
Author Response
I think that the error "in the experiments in this study" [not (only) the theoretical and analytical error but the experimental error] should be comprehensively investigated. Then, it is better to show concretely what error occurred and how much. Moreover, it is better to clarify the following points.
- A0 mode velocity depends on the frequency, and therefore, if the output frequency is different from the input frequency or if the frequency bandwidth exists, will there be an error or increased uncertainty?
For measurements we used short wide band ultrasonic pulses and the frequency 50 kHz corresponds to the central frequency of the electric pulse used for excitation of the ultrasonic transducer. In the PVC film propagates an ultrasonic pulse the spectrum of which is shown in Figure 9,b. Due to the frequency response of the ultrasonic transducer with the matching layer the maximum is observed at 52 kHz. Therefore, for thickness measurements the frequency 52 kHz was used. Yes, one of the uncertainty components of the A0 mode phase velocity measurements may be due to the frequency deviation. However, the influence of the frequency deviation on the thickness measurement is by one order smaller than the influence of the measurement uncertainty of the S0 mode phase velocity. Change of the frequency by 2 kHz (from 50 to 52 kHz) causes 0.1% change in the film thickness measurement for a 200 µm thick film (Equation 8) and only 0.07% for a 150 µm thick film.
This explanation was inserted in the manuscript.
- Why is the uncertainty about the measured value of the 200 µm film about twice as large as that of the 150 µm film? (Is it possible to less accurately measure in principle when the film is thicker?)
The uncertainty due to the measured 200 µm film value is approximately twice that of the 150 µm film due to the higher slope of the A0 mode dispersion curve for the thicker film in the same frequency range (Figure 3). This explanation was inserted in the manuscript.
- Why are both measured values of 150 and 200 µm films smaller than the measured value using a micrometer?
It seems that the manufacturer always is declaring a bigger thickness of the film than it is in reality. That was confirmed by a lot of measurements performed by the micrometer. We also observed thickness variations over the whole area of the film up to 25 µm. Meanwhile, the film thickness values measured by the micrometer and ultrasound are close to each other after estimating possible measurement uncertainties (Table 2).
Given that the uncertainty is a few percent, isn't three significant digits reasonable?
The values of the film thickness are corrected in the edited manuscript.
For "PVC" in the abstract, it should be specified what the abbreviation is.
The abbreviation was inserted into the edited manuscript.